# Longitudinal Trajectories of Alcohol Consumption with All-Cause Mortality, Hypertension, and Blood Pressure Change: Results from CHNS Cohort, 1993–2015

**DOI:** 10.3390/nu14235073

**Published:** 2022-11-29

**Authors:** Weida Qiu, Anping Cai, Liwen Li, Yingqing Feng

**Affiliations:** 1The Second School of Clinical Medicine, Southern Medical University, Guangzhou 510515, China; 2Department of Cardiology, Hypertension Research Laboratory, Guangdong Cardiovascular Institute, Guangdong Provincial People’s Hospital, Guangdong Academy of Medical Sciences, Guangzhou 510100, China

**Keywords:** alcohol consumption, trajectory, all-cause mortality, hypertension, blood pressure

## Abstract

Background: Previous studies have demonstrated a J-shaped association of alcohol consumption with all-cause mortality and hypertension, but the majority of these studies focus on a single measurement of alcohol intake and were conducted in a Western population. Whether long-term trajectories of alcohol consumption are associated with all-cause mortality, hypertension, and a change in blood pressure remains to be elucidated. Methods: In the large, population-based China Health and Nutrition Survey cohort from between 1993 and 2015, group-based trajectory modeling was conducted to identify distinct alcohol-consumption trajectory classes. We investigated their association with all-cause mortality and hypertension using Cox regression and binary logistics regression models. A restricted cubic spline was performed to determine the nonlinear relationships of mean alcohol intake with mortality and hypertension. Multivariate-adjusted generalized linear mixed-effects models were conducted to assess the change in blood pressure among alcohol-consumption trajectory classes. Results: Among the 5298 participants, 48.4% were women and the mean age was 62.6 years. After 22 years of follow-up, 568 (10.7%) of the participants died and 1284 (24.2%) developed hypertension. Long-term light and moderate drinkers had a lower risk of death than the non-drinkers, and a restricted cubic spline showed a J-shaped relationship between mean alcohol intake and mortality. Although blood pressure increased slower in light and moderate drinkers, a reduced risk of hypertension was only observed in the former. The long-term heavy drinkers had the highest blood pressure and death rate. Conclusions: Light alcohol intake might be protective even in the long run, while heavy drinking reversed the beneficial effect. The causality of such a connection needs to be further investigated.

## 1. Introduction

Alcohol is a recognized risk factor for mortality and several chronic diseases, including hypertension, diabetes, and cancers [1,2,3], as well as mental illnesses and other alcohol-use disorders [4]. It was reported that ethanol causes over 3 million deaths per year and present a heavy total burden of injury and diseases worldwide by WHO in 2018 [5]. Although the trend is decreasing globally, alcohol consumption increases at a terrific speed in China with a 76% increment, relative to 2005.

Similarly, hypertension also causes an enormous social burden and growing mortality around the world [6]. Furthermore, there is still a steady increase in the prevalence of elevated blood pressure (BP) from pole to pole [7]. Hence, hypertension and alcohol consumption have been listed as two of the five risk factors for non-communicable diseases [8]. Previous studies have demonstrated the relationship between alcohol consumption and hypertension [9] or mortality [10], but these studies only focus on a single measurement and were conducted in non-Chinese population. However, drinking patterns and intake can change over time and they vary in different countries [5]; therefore, detecting a distinct trajectory of longitudinal change in alcohol consumption can yield a better understanding of the association between long-term cumulative alcohol impact and adverse outcomes (i.e., hypertension, all-cause mortality). Notably, using a single measurement of alcohol intake might confuse a current drinker with a former drinker [11], such as in a situation in which someone drinks at baseline and quits drinking during follow-up.

To the best of our knowledge, no study has investigated the associations between a long-term alcohol-consumption trajectory and all-cause mortality, hypertension, and a change in BP using group-based trajectory modeling (GBTM) among Chinese adults. To fill this knowledge gap, we identified a distinct alcohol intake trajectory over a 22-year period and assessed these associations by leveraging the China Health and Nutrition Survey (CHNS) cohort.

## 2. Methods

### 2.1. Study Design and Participants

Details of the ongoing, community-based CHNS cohort are described elsewhere [12,13,14]. Briefly, the CHNS data were collected from 12 provinces through a multistage, random cluster-sampling process. One large city and another small city (usually the provincial capital and a lower-income city), as well as four counties (one high-, one low-, and two middle-income counties), in each province were selected using this sampling strategy. Within each city, two urban and two rural communities were randomly selected. Within each county, one community from the capital city and three villages from rural areas were randomly chosen. In each community, twenty households were randomly selected. Based on the diversity of geography, economic development, public resources, and health indicators, these data can be used as a representative dataset for all provinces in China. Study protocols were approved by the Institutional Review Committees of the University of North Carolina at Chapel Hill, USA and the National Institute for Nutrition and Health at the Chinese Center for Disease Control and Prevention in Beijing, China. All participants provided written informed consent and the study was conducted in accordance with the Declaration of Helsinki.

The CHNS was initiated in 1989 and has been followed up every two to four years in 1991, 1993, 1997, 2000, 2004, 2006, 2009, 2011, and 2015 among more than 30,000 individuals. Given that alcohol-consumption data were collected from 1993, the participants in the current study were selected from 1993 to 2015. We first included participants with at least three alcohol intake records with at least one from 1993. Those who were <18 years of age in 1993 and diagnosed with hypertension before 1993 were also excluded. Finally, a total of 5298 participants were recruited in the final study (Appendix A).

### 2.2. Alcohol-Consumption Measurement

Information on drinking behaviors was collected using face-to-face surveys at each follow-up round from 1993. Participants were asked whether they had ever drunk beer, liquor, or other alcoholic beverages in the last year. If the answer was yes, the alcohol drinkers were further asked whether they drunk alcohol weekly (i.e., bottles per week for beer, liang per week for wine and liquor). Alcohol concentration was in line with the 2010 China monitoring report on chronic disease risk factors [15]: beer = 4%; wine = 10%; and liqueur = 38% (1 bottle = 600 mL, 1 Liang = 50 mL). The formulas used to estimate the total intake of ethanol consumed are as follows:(1)Beer(grams)=bottle∗600 mL∗4%
(2)Wine (grams)=liang∗50 mL∗10%
(3)Liquor (grams)=liang∗50 mL∗38%

Finally, alcohol consumption was converted into UK units [16], namely 1 unit equivalent to 8 g of ethanol.

### 2.3. Outcome Identification

The study outcomes of interest were all-cause mortality, newly onset hypertension, and BP change. Face-to-face surveys with participants and their family members were used to capture all-cause mortality and newly onset hypertension events at each follow-up wave. The exact date of death was ascertained by family members during follow-up surveys, and the date of death or the last survey date was used to calculate the follow-up time, whichever came first. In addition, newly onset hypertension was defined as the current use of anti-hypertensive treatment or a diagnosis by a physician.

Seated BP was measured on the participants’ right arm by experienced physicians using standard mercury sphygmomanometers at each visit. Following standard protocol, participants were asked to rest and sit quietly for 15 min and avoid smoking and caffeine intake at least 1 h before measurement. The BP of each participant was measured 3 times with a 30 s interval. The mean value of the 3 measurements at each visit was used. Systolic blood pressure (SBP) and diastolic blood pressure (DBP) were determined using the first and fifth phases of the Korotkoff method, and pulse pressure was defined as the difference between SBP and DBP.

### 2.4. Covariates

Age was calculated as the year of the last survey or death minus the year of birth. Other demographic information, including gender, marital status (married or others), residence area (urban or rural), educational level (less than high school, high school or equivalent, or college or above), smoking status (current, former, or never), and previous history of diseases, including diabetes mellitus (yes or no), stroke (yes or no), myocardial infraction (MI) (yes or no), and malignant tumor (yes or no), were collected from the last survey. Mean BP, mean body mass index (BMI), mean waist circumference, and mean alcohol consumption were calculated as the average value of measurements from 1993 to 2015.

### 2.5. Statistical Analysis

To detect distinct trajectories of longitudinal alcohol-consumption development, GBTM was performed with the ‘traj’ command [17] in STATA. A detailed description of the creation of alcohol-consumption trajectories is presented in the Appendix A. Briefly, participants who reported not drinking in each survey were categorized as non-drinkers and were excluded from GBTM analysis. GBTM, a specialized form of finite mixture modeling, was then performed to identify groups of individuals who shared similar alcohol intake trajectories over time [18]. Model accuracy was based on standard criteria, including the Bayesian information criterion (BIC) and the posterior probability [18,19]. We also maintained the average of the posterior probability of the assignments above 0.7 in all cases and group sizes above 5% of participants, which are regarded as acceptable [18].

A final three-trajectory group with the “5,2,4” model (N = 3060) for alcohol consumption was chosen (Figure 1): (i) light drinker (N = 1712, 56.0%), alcohol consumption was maintained at a light level (less than 10 units per week in the long term); (ii) moderate drinker (N = 1169, 38.2%), alcohol consumption was maintained at a moderate level (approximately 20 units per week); (iii) heavy drinker (N = 179, 5.9%), alcohol consumption was maintained at a high level for a long time (more than 40 units per week) before decreasing, but ultimately it remained the highest level. Finally, 5298 participants were divided into 4 classes: non-drinker; light drinker; moderate drinker; and heavy drinker.

The participants’ characteristics are presented as mean ± SD or median and interquartile range for the continuous variables and proportions for the categorical variables. Differences in baseline characteristics between trajectory groups were tested using one-way ANOVA, Kruskal–Wallis H-tests, and χ^2^-tests, accordingly. Cox regression analysis and binary logistics regression analysis were performed to estimate the cumulative rate of all-cause mortality and the association between newly onset hypertension with different alcohol-consumption trajectory groups. The logistics model was adjusted for age, sex, marital status, education, residence area, smoking status, mean BMI, and mean waist circumference with a stepwise forward adjustment, and the Cox regression model was adjusted for the above covariates as well as self-reported comorbidities. To determine whether there was a nonlinear association of alcohol consumption with all-cause mortality and hypertension, a restricted cubic spline was performed, adjusting for the same covariates used in the logistics model, and Akaike’s information criterion was used to determine the optimal number of knots [20].

To assess the relationship between the change in BP and alcohol-consumption classes, multivariate-adjusted generalized linear mixed-effects models were conducted, with an adjustment for the identical variates included in the logistics regression model. Histograms of residuals and residual plots were shown to confirm the normal distribution of residuals. Multiple comparisons were carried out in the linear mixed-effects models, with the significance level set to *p* < 0.017 (0.05÷3 comparisons = 0.017). Two-sided *p* values < 0.05 were considered statistically significant for other analyses. All analyses were carried out with Stata 15.0 (StataCorp, College Station, TX, USA) and R (version 4.1.2; The R Foundation for Statistical Computing, Vienna, Austria).

## 3. Results

### 3.1. Characteristics of the Study Population

As illustrated in the study flow chart (Appendix A), a total of 5298 subjects were included in the current study. Based on the distinct longitudinal alcohol intake, participants were categorized into four classes: non-drinker; light drinker; moderate drinker; and heavy drinker (Figure 1).

The baseline characteristics of the participants according to different alcohol-consumption trajectories are presented in Table 1. Among the study participants, 48.4% were men, the mean age was 62.6 years, and the median alcohol consumption was 0.75 (0–8.8) units per week. Among them, the non-drinkers were more likely to be older, unmarried, and have an education level less than high school. They were also mostly comprised of men and non-smokers. The prevalence of diabetes mellitus in non-drinkers tended to be higher than the moderate drinkers and heavy drinkers. The heavy drinkers were more likely to live in rural areas, be current and former smokers, and have a higher mean SBP and DBP. The prevalence of MI, stroke, and tumor was comparable among the four classes.

### 3.2. Alcohol Consumption and All-Cause Mortality

During a median follow-up of 22.0 years (97,742 person-years), 568 (10.7%) of the participants died. When longitudinal alcohol consumption was categorized into four classes (non-drinker, light drinker, moderate drinker, and heavy drinker), the cumulative rate of all-cause mortality was higher in the heavy drinker group compared to the non-drinker group (crude hazard ratio (HR): 1.55; 95% confidence interval (CI):1.07, 2.25) in the univariate Cox regression model. While further adjusting for the covariates, the light drinkers and moderate drinkers had a 24% (HR: 0.76; 95% CI: 0.61, 0.95) and 39% (HR: 0.61; 95% CI: 0.46, 0.81) decreased risk of death relative to the non-drinkers (Table 2). The restricted cubic spline showed a J-shaped relationship between mean alcohol consumption and all-cause mortality (*p* for nonlinear < 0.001), with a risk nadir of approximately 10–20 units per week (Figure 2, Panel A).

### 3.3. Alcohol Consumption and Newly Onset Hypertension

A total of 1284 (24.2%) participants developed newly onset hypertension after a 22-year follow-up. Alcohol-consumption trajectories were not significantly related to newly onset hypertension in the univariate binary logistics regression model. While further adjusting for the covariates, light drinker had a 16% (odds ratio (OR): 0.84; 95% CI: 0.72, 0.99) decreased risk of hypertension compared to non-drinkers (Table 3). Among them, the restricted cubic spline showed a non-linear relationship (*p* for nonlinear = 0.001) between mean alcohol consumption and newly onset hypertension (Figure 2 Panel B).

### 3.4. Alcohol Consumption and Change in Blood Pressure

For SBP, the light drinkers (Slope: −0.17 (−0.04, −0.30)) and moderate drinkers (Slope: −0.19 (−0.32, −0.06)) had a lower speed of BP increase than the heavy drinkers. This relationship was also observed when comparing with the non-drinkers, while the speed of SBP increase was comparable between the non-drinkers and heavy drinkers (Figure 3).

For DBP, the difference in the speed of increase was only observed between the light drinkers and moderate drinkers (Slope: 0.06 (0.02, 0.11); relative to light drinkers) (Appendix A).

For PP, moderate drinkers had the lowest speed of increase among the four classes (slope: −0.18 (−0.29, −0.08), relative to heavy drinkers; slope: −0.06 (−0.11, −0.01), relative to light drinkers; slope: −0.13 (−0.18, −0.08), relative to non-drinkers). In addition, the PP of non-drinkers elevated faster than that of the light drinkers (slope: 0.07 (0.02, 0.11)) (Figure 4).

The histogram of residuals and the residual plots for SBP, DBP, and PP proved to have a normally distributed fit (Appendix A).

## 4. Discussion

In this large prospective population-based cohort study, covering 12 provinces in Mainland China from 1993 to 2015, long-term light drinkers had a lower risk of mortality and hypertension, and their SBP and PP increased more rapidly than the non-drinkers. The reduced risk was only observed in all-cause mortality and not in newly onset hypertension, although the SBP and PP in long-term moderate drinkers increased more slowly than the non-drinkers. Among them, the long-term heavy drinkers had the highest BP and death rate.

Numerous studies have demonstrated that modest alcohol consumption is associated with a lower risk of all-cause mortality. In a large U.S. cohort, Xi and colleagues emphasized the J-shaped relationship between alcohol intake and all-cause mortality, suggesting that light-to-moderate drinking might be protective [21]. Among the high-income European population, Angela M Wood et al. found the lowest threshold for all-cause mortality was about 100 g/week [22]. Even in the populations who survived an initial MI, the beneficial effect on mortality was also observed in those with a modest alcohol intake [23]. However, most of these studies focused on a single record of alcohol intake, which might ignore the change in drinking patterns and the cumulative effect of ethanol [24] on health in the long run. Our present results support and extend the prior findings. Among a Chinese population, we categorized four alcohol consumption classes using GBTM with a 22-year follow-up period and found that long-term light drinkers (less than 10 units/week) and moderate drinkers (10–20 units/week) had a lower risk of death than abstainers. Furthermore, using a 22-year record of ethanol intake to calculate mean alcohol consumption, a J-shaped relationship with all-cause mortality was observed.

However, whether a protective effect from modest alcohol intake exists has been debated intensively. According to a meta-analysis involving seven high-quality cohort studies, moderate alcohol consumption does not lower mortality risk [25], and another pooled analysis of nine national populations from the European Union showed that the protective association between light-to-moderate alcohol intake and all-cause mortality was only observed in women ≥ 65 years of age [26]. Furthermore, moderate alcohol consumption was found to not be causally correlated with stroke prevention in a recent genetic epidemiology study among a Chinese cohort [27]. Moreover, the J-shaped association between alcohol consumption and coronary heart disease risk was not confirmed by previous Mendelian randomization analysis [28], which is less susceptible to confounding and reverse causality than observational studies. Hence, researchers believed that the beneficial effect of modest alcohol intake was mainly due to selection bias (i.e., non-drinkers are more likely to be sick) and an inadequate adjustment for covariates [26,28]. Nevertheless, the prevalence of MI, stroke, and cancer, which have an important impact on survival, was comparable between long-term abstainers and drinkers, and our conclusions were drawn after adjusting for multiple self-reported comorbidities. We acknowledge that many confounders might be ignored in our current study and have taken the inconsistent previous results and carcinogenic effects of ethanol [22,29] into consideration. We do not encourage non-drinkers to begin drinking, but advocate the idea of a drinking constraint for heavy drinkers and binge drinkers. This important question will require further quasi-experimental studies to be addressed in the future.

Consistent with previous publications, our study demonstrates the lower speed of SBP and PP increase among light drinkers and moderate drinkers compared to non-drinkers and heavy drinkers, indicating a vasculo-protective effect against arterial stiffening in subjects without hypertension [30]. Previous studies have confirmed the beneficial effect of light-to-moderate alcohol consumption on endothelial function, while heavy alcohol intake played a pathogenic role in animal models [31,32]. SBP and PP, rather than PP, increased with the development of arterial stiffness, and the findings of our study were indirectly in line with the previous study viewpoint that moderate alcohol consumption was associated with a lower risk of atherosclerosis [33]. Despite the lower BP increase rate observed in moderate drinkers, only the light drinkers had a reduced risk of hypertension in the present study. In addition, the restricted cubic spline further confirmed a curvilinear relationship between alcohol intake and hypertension with a risk nadir of 0–10 units/week in our study. However, same as the relationship between alcohol intake and mortality, the association between alcohol consumption and hypertension is inconclusive [34], and whether to drink moderately or to not drink should be preferred as a non-pharmacological means of BP control remains to be elucidated [35,36]. However, one thing is conclusive and convincing, that is, reducing alcohol intake for heavy drinkers makes sense [37].

Nowadays, abstinence may be always driven by illness rather than subjective ideas [38], which can be reflected in the higher comorbidity rates of abstainers [21]. This has resulted in a paradox, that is, former drinkers have a poor prognosis relative to moderate drinkers [11]. Except for individuals’ health, alcohol withdrawal syndrome is contributed to poor prognosis as well [39]. Indeed, alcohol withdrawal is associated with injury and mental illness. Taking our present results into consideration, being a light drinker may be a better option or transitive bridge for those drinking heavily.

Our study may have benefited from the following strengths. Using a longitudinal, population-based sampling strategy, the CHNS data provided a unique opportunity to analyze the long-term alcohol-consumption trajectory among China and further investigate the relationship between all-cause mortality and BP. To our knowledge, our study is the first to investigate alcohol intake trajectory and clinical outcomes using GBTM in a Chinese population.

Nevertheless, several important limitations should be noted. First, the CHNS is not a nationally representative population, and the results must be interpreted with caution for generalizability. Second, the observational study has its inherent limitations, including the collection of non-randomized data, missing potential confounding factors, and causality cannot be proved. Third, alcohol consumption was self-reported, which may underestimate alcohol intake, and information on the type of alcoholic beverages consumed was not considered. However, a previous study proved that self-reported measures of ethanol consumption were reliable in moderate drinkers [40], and another meta-analysis demonstrated that moderate drinkers with all alcoholic drinks were related to a lower heart disease risk, suggesting that the major effect is from ethanol instead of other components of alcoholic beverages [41]. Fourth, GBTM analysis has weaknesses as well. For instance, trajectory classes created by GBTM have very different sizes, making comparison between subgroups difficult due to the low statistical power. Nevertheless, because drinking intake is not static and changes with time, GBTM analysis can cluster participants more flexibly based on heterogeneous growth patterns.

## 5. Conclusions

Using a population-based, longitudinal cohort of Chinese adults, our study found curvilinear curves in alcohol–mortality and alcohol–hypertension associations using average alcohol consumption during a 22-year period, extending previous findings that light alcohol intake might be protective in the long run, while heavy drinking reverses the beneficial effect. The causality of such a connection needs to be further investigated, but the reduction in the high consumption of alcohol should always be bore in everyone’s mind.

## 6. Novelty and Relevance

### 6.1. What Is New?

Despite controversy, most of the previous studies have demonstrated a J-shaped relationship between alcohol consumption and adverse outcomes. However, the majorities of research focuses on a single measurement and were conducted in non-Chinese populations. Herein, we investigated the association of longitudinal alcohol intake trajectories, classed by group-based trajectory modeling, with all-cause mortality and blood pressure using a large population-based cohort in China. The curvilinear curves in the alcohol–mortality and alcohol–hypertension associations were re-emphasized using average alcohol consumption during a 22-year period, extending previous findings that light alcohol intake might be protective in the long run, and supporting the notion that heavy drinking reverses this beneficial effect.

### 6.2. What Is Relevant?

Alcohol consumption is a major risk factor for mortality and hypertension, and alcohol problems are particularly acute in China. As such, exploring potential long-term drinking intake to optimize public health management is of great importance. In this large population-based Chinese cohort, we identified four classes of alcohol-consumption trajectories and found that long-term light drinking had the lowest risk of mortality and hypertension. The curvilinear associations of mean alcohol intake with mortality and hypertension were also observed.

## 7. Clinical/Pathophysiological Implications?

Although lower risks of death and hypertension were observed among light drinkers, the causality of such a connection needs to be further investigated. Additionally, whether to drink moderately or to not drink is optimal remains to be elucidated, and future studies are needed to determine whether alcohol withdrawal from a moderate intake to zero in a healthy population is beneficial. For heavy drinkers, a reduction in high alcohol consumption is essential.

## Figures and Tables

**Figure 1 nutrients-14-05073-f001:**
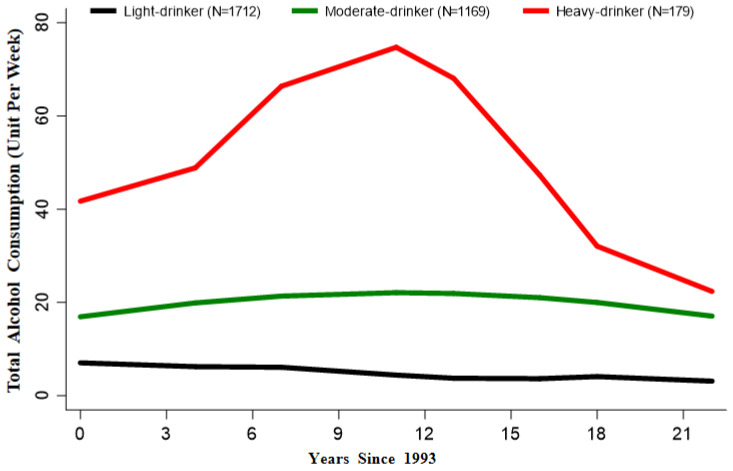
Trajectories from 1993 to 2015 for alcohol consumption.

**Figure 2 nutrients-14-05073-f002:**
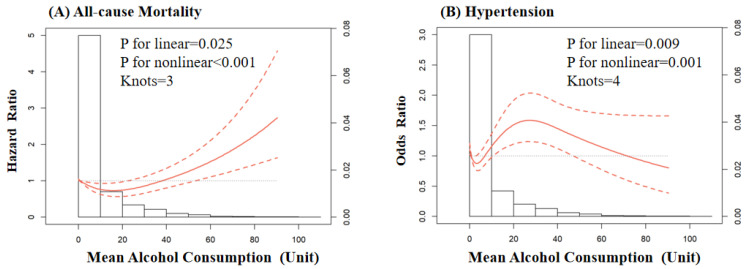
Curvilinear association of mean alcohol consumption with all-cause mortality (**A**) and newly onset hypertension (**B**). Panel (**A**) was conducted with restricted cubic spline using Cox regression model. Panel (**B**) was conducted with restricted cubic spline using logistics regression model. Both panels (**A**,**B**) were adjusted for age, sex, marital status, education, residence area, smoking status, mean BMI, mean waist circumference.

**Figure 3 nutrients-14-05073-f003:**
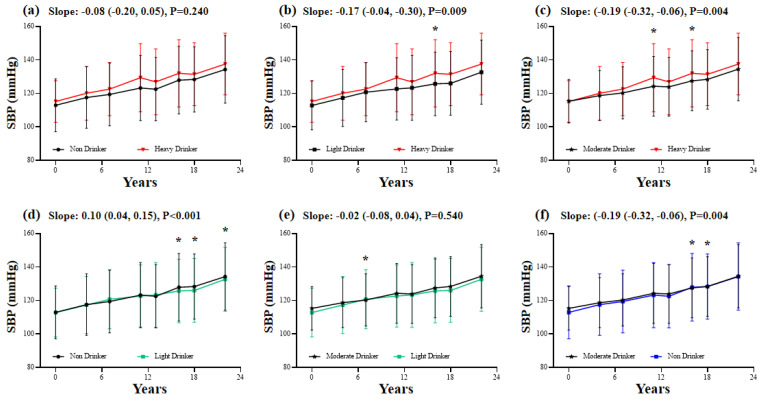
The association between alcohol consumption and change in systolic blood pressure. The multivariate-adjusted generalized linear mixed-effects models were adjusted for age, sex, marital status, education, residence area, smoking status, mean BMI, and mean waist circumference. The first group with a black color at each panel was the reference group. *p*-value < 0.017 was considered statistically significant for the multiple comparisons. * Blood pressure was significantly different between groups.

**Figure 4 nutrients-14-05073-f004:**
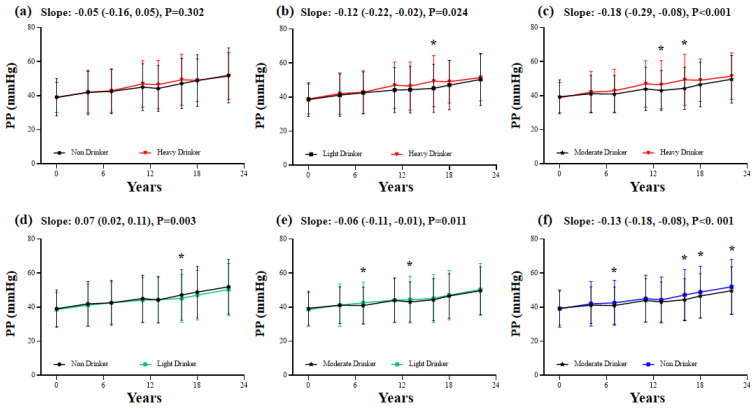
The association between alcohol consumption and change in pulse pressure. The multivariate-adjusted generalized linear mixed-effects models were adjusted for age, sex, marital status, education, residence area, smoking status, mean BMI, and mean waist circumference. The first group with a black color at each panel was the reference group. *p*-value < 0.017 was considered statistically significant for the multiple comparisons. * Blood pressure was significantly different between groups.

**Table 1 nutrients-14-05073-t001:** Baseline characteristics according to different alcohol-consumption trajectory classes.

Variables	Overall (N = 5298)	Non-Drinker (N = 2238)	Light Drinker (N = 1712)	Moderate Drinker (N = 1169)	Heavy Drinker (N = 179)	*p*-Value
Demographics						
Age (years)	62.6 ± 12.7	64.1 ± 13.0	62.4 ± 13.0	60.1 ± 11.4	63.3 ± 9.8	<0.001
Male, (n%)	2564 (48.4)	277 (12.4)	994 (58.1)	1115 (95.4)	178 (99.4)	<0.001
Married, n (%)	4310 (81.4)	1695 (75.7)	1399 (81.7)	1055 (90.3)	161 (89.9)	<0.001
Urban, n (%)	1396 (26.4)	550 (24.6)	488 (28.5)	321 (27.5)	37 (20.7)	0.010
Education						
Less than high school, n (%)	4369 (82.5)	1942 (86.8)	1383 (80.8)	899 (76.9)	145 (81.0)	<0.001
High school or equivalent, n (%)	782 (14.8)	259 (11.6)	270 (15.8)	223 (19.1)	30 (16.8)	
College or above, n (%)	147 (2.8)	37 (1.7)	s59 (3.5)	47 (4.0)	4 (2.2)	
Physical examination *						
Mean SBP (mmHg)	121.9 ± 13.3	121.7 ± 14.2	121.4 ± 13.2	122.7 ± 11.8	125.2 ± 12.6	<0.001
Mean DBP (mmHg)	78.3 ± 7.5	77.4 ± 7.6	78.0 ± 7.4	79.8 ± 7.2	80.4 ± 7.8	<0.001
Mean PP (mmHg)	43.7 ± 8.7	44.3 ± 9.4	43.4 ± 8.6	42.9 ± 7.5	44.8 ± 8.5	<0.001
Mean BMI (kg/m^2^) †	22.5 (20.7–24.5)	22.5 (20.7–24.8)	22.4 (20.8–24.4)	22.4 (20.8–24.5)	21.9 (20.6–23.3)	0.047
Mean waist circumference (cm) †	79.3 (74.3–85.0)	78.7 (73.4–84.4)	79.3 (74.5–85.2)	80.5 (75.5–86.3)	79.8 (75.1–84.8)	<0.001
Smoking status						
Never, n (%)	2800 (84.9)	1900 (84.9)	767 (44.8)	121 (10.4)	12 (6.7)	<0.001
Former, n (%)	459 (8.7)	82 (3.7)	204 (11.9)	146 (12.5)	27 (15.1)	
Current, n (%)	2039 (38.5)	256 (11.4)	741 (43.3)	902 (77.2)	140 (78.2)	
Alcohol consumption *						
Mean alcohol consumption unit †	0.75 (0–8.8)	0	2.2 (0.8–4.9)	17.2 (11.5–25.6)	51.8 (42.7–61.3)	<0.001
Self-reported comorbidities						
Hypertension, n (%)	1284 (24.2)	581 (26.0)	378 (22.1)	282 (24.1)	43 (24.0)	0.047
Diabetes mellitus, n (%)	301 (5.7)	147 (6.6)	102 (6.0)	46 (3.9)	6 (3.4)	0.007
Myocardial infraction, n (%)	124 (2.3)	53 (2.4)	50 (2.9)	17 (1.5)	4 (2.2)	0.077
Stroke, n (%)	182 (3.4)	71 (3.2)	69 (4.0)	39 (3.3)	3 (1.7)	0.262
Malignant tumor, n (%)	40 (0.8)	14 (0.6)	16 (0.9)	8 (0.7)	2 (1.1)	0.543

* Mean values were calculated using data from 1993, 1997, 2000, 2004, 2006, 2009, 2011, and 2015. † Presented as median (interquartile range).

**Table 2 nutrients-14-05073-t002:** The association between alcohol-consumption trajectory classes and all-cause mortality.

	Events/Total (%)	Person-Years	Crude HR (95% CI)	*p*-Value	Adjusted HR * (95% CI)	*p*-Value
All participants	568/5298 (10.7%)	97,742	-	-	-	-
Non-drinker	241/2238 (10.8)	40,728	Ref	Ref	Ref	Ref
Light drinker	185/1712 (10.8)	31,800	0.97 (0.80–1.18)	0.771	0.76 (0.61–0.95)	0.014
Moderate drinker	111/1169 (9.5)	21,878	0.84 (0.67–1.06)	0.136	0.61 (0.46–0.81)	0.001
Heavy drinker	31/179 (17.3)	3337	1.55 (1.07–2.25)	0.022	0.95 (0.63–1.43)	0.792

* Adjusted for age, sex, marital status, education, residence area, smoking status, mean BMI, mean waist circumference, self-reported hypertension, diabetes mellitus, myocardial infraction, stroke, and malignant tumor in multivariate Cox regression model.

**Table 3 nutrients-14-05073-t003:** The association between alcohol-consumption trajectory classes and newly onset hypertension.

	Events/Total (%)	Crude OR (95% CI)	*p*-Value	Adjusted OR * (95% CI)	*p*-Value
All participants	1284/5298 (24.2)	-	-	-	-
Non-drinker	581/2238 (26.0)	Ref	Ref	Ref	Ref
Light drinker	378/1712 (22.1)	0.81 (0.70–0.94)	0.771	0.84 (0.72–0.99)	0.038
Moderate drinker	282/1169 (24.1)	0.91 (0.77–1.07)	0.242	1.13 (0.94–1.35)	0.187
Heavy drinker	43/179 (24.0)	0.90 (0.63–1.29)	0.569	1.08 (0.74–1.57)	0.692

* Adjusted for age, sex, marital status, education, residence area, smoking status, mean BMI, mean waist circumference in multivariate binary logistics regression model.

## Data Availability

The data analyzed in this study are available on request from the CHNS research group (https://www.cpc.unc.edu/projects/china, accessed on 29 August 2022).

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
