# Peer review of "Longitudinal Trajectories of Alcohol Consumption with All-Cause Mortality, Hypertension, and Blood Pressure Change: Results from CHNS Cohort, 1993–2015"

_nutrients, 2022, doi:10.3390/nu14235073_

Round 1

Reviewer 1 Report

The reviewed work is original. The study confirms that the consumption of moderate to large amounts of alcohol is harmful to health and leads to death.

Another limitation of the study is that the study was conducted for increased blood pressure, hypertension or mortality. What other health effects, for example, liver disease, social effects, have not been studied.

Author Response

Thanks for the valuable comments. Indeed, the relationships between alcohol intake and other health effects (i.e., liver dysfunction, mental disease, social effects) are of great importance and have been received extensive attention. However, the CHNS did not collect the data on mental and liver diseases and social effects, so the current study was mainly conducted for blood pressure progression, incident of hypertension or mortality, which are the significant public health issues, and their associations with long-term alcohol consumption trajectories remain to be elucidated.

Reviewer 2 Report

Thank you for the opportunity to review this paper which reports the trajectories of alcohol consumption with all-cause mortality, hypertension and blood pressure change. I have some major and minor comments, particularly around the finding of the J-shaped curve which the authors do discuss.

Major comments:

1. The finding of the J-shaped curve regarding alcohol consumption and all-cause mortality and hypertension is based on a measure where participants were asked if they had ever drank alcohol in the past year but later in the discussion there is mention of non-drinkers being lifetime abstainers (line 222). However, it is not possible to know whether a participant had ever drank alcohol if the measure was based on past year alcohol consumption.

With this in mind, the authors should be wary of some of the conclusions around the findings given that it does not seem that they have looked at lifetime abstinence rather non-drinking in the past year.

2. I appreciate that there may be word limits to submission, but if the sample is not representative of the general population, then the authors should describe the design and sampling of the CHNS in more detail to make it clear whether participants lived in private households, etc.

Minor comments:

3. Gender was mentioned in the statistical analysis section but not the covariates section, please add this here.

4. Please provide justification for the cut-offs used to define light, moderate and heavy drinkers.

5. There were some minor grammar and spelling issues throughout the manuscript.
